# Comprehensive Analysis of Transcriptome and Metabolome Reveals the Flavonoid Metabolic Pathway Is Associated with Fruit Peel Coloration of Melon

**DOI:** 10.3390/molecules26092830

**Published:** 2021-05-10

**Authors:** Aiai Zhang, Jing Zheng, Xuemiao Chen, Xueyin Shi, Huaisong Wang, Qiushi Fu

**Affiliations:** Key Laboratory of Biology and Genetic Improvement of Horticultural Crops of the Ministry of Agriculture and Rural Affairs, Institute of Vegetables and Flowers, Chinese Academy of Agricultural Sciences, Beijing 100081, China; aiq1998@163.com (A.Z.); Zj34312021@163.com (J.Z.); a19801167202@163.com (X.C.); shixueyin123@163.com (X.S.)

**Keywords:** melon, fruit peel color, transcriptome, metabolome, flavonoid biosynthesis

## Abstract

The peel color is an important external quality of melon fruit. To explore the mechanisms of melon peel color formation, we performed an integrated analysis of transcriptome and metabolome with three different fruit peel samples (grey-green ‘W’, dark-green ‘B’, and yellow ‘H’). A total of 40 differentially expressed flavonoids were identified. Integrated transcriptomic and metabolomic analyses revealed that flavonoid biosynthesis was associated with the fruit peel coloration of melon. Twelve differentially expressed genes regulated flavonoids synthesis. Among them, nine (two *4CL*, *F3H*, three *F3′H*, *IFS*, *FNS*, and *FLS*) up-regulated genes were involved in the accumulation of flavones, flavanones, flavonols, and isoflavones, and three (2 *ANS* and *UFGT*) down-regulated genes were involved in the accumulation of anthocyanins. This study laid a foundation to understand the molecular mechanisms of melon peel coloration by exploring valuable genes and metabolites.

## 1. Introduction

Melon (*Cucumis melo* L.) belongs to the Cucurbitaceae family, and it is one of the most important economic crops worldwide [1]. Melon peel is essentially the boundary of the fruit. It not only maintains the integrity of the fruit but also protects it from the external environment. Peel color is one of the important characteristics that affect the commercial value of melon fruit and consumer choice [2]. Peel color variety of melon fruit includes white, yellow, orange, and green [3]. Melon peels are rich in nutritional ingredients, such as carbohydrates and minerals, and they contain significant amounts of dietary fibers and antioxidants, such as polyphenols and flavonoids [4].

Flavonoids are a large class of polyphenolic secondary metabolites. There are mainly six groups of flavonoids in plant tissues, including flavones, flavanones, isoflavones, anthocyanins, flavanols, and flavonols [5]. Most flavonoids are identified as antioxidants, protecting plants against various stresses [2,6]. In addition, flavonoids are the pigments that affect the coloration of plant organs [7]. In humans, flavonoids play important roles in maintaining normal vascular permeability and protecting against diseases, such as hyperglycemia, cancer, and diabetes [8].

In recent years, high-throughput methods have been widely applied to understand pigmentation in plants. By comprehensive analysis of the metabolome and transcriptome, the relationship between the content of various secondary metabolites and the corresponding differentially expressed genes has been revealed. Some structural genes and transcription factors were found to be involved in flavonoid biosynthesis. Three UDP-glucose flavonoid 3-*O*-glucosyltransferase genes (*UFGT*) are involved in the accumulation of malvidin 3-*O*-glucoside and delphinidin 3-*O*-glucoside, which correlate with jujube peel coloration [9]. Another important study in strawberry identified three genes, chalcone synthase (*CHS*), dihydroflavonol reductase (*DFR*), and flavonol 3-*O*-glucosyltransferase (*F3H*), as well as three anthocyanin metabolites, cyanidin 3-*O*-glucoside chloride, cyanidin 3-galactoside, and cyanidin 3-glucoside, that are involved in the red fruit color [10]. Combining transcriptome and metabolome approaches, Zhang et al. [11] revealed that the red mutation cultivar ‘Red Clapp Favorite’ of pear was caused by the *PcGSTF12,* which was involved in anthocyanin and procyanidin accumulation. Recently, γ-carotene, *CRTISO*, and *ε-LCY* were identified as candidates involved in the orange pigmentation in *Liriodendron tulipifera* [12].

Little is known about the regulation of flavonoid biosynthesis in Cucurbitaceae. Flavonoids are accumulated in leaves and the reproductive organs of some cucurbits of cucumber and muskmelon [13]. Tadmor et al. [3] found that melon rind color is based on different combinations of chlorophyll, carotenoids, and flavonoids according to the cultivar tested. In addition, Gur et al. [14] reported that a kelch domain-containing F-box protein-coding gene (*CmKFB*) regulates naringenin chalcone accumulation in muskmelon, resulting in peel yellowing. Interestingly, this result perfectly matches the conclusions of Feder et al. [15]. Recently, Oren et al. [16] also identified an arabidopsis pseudo-response regulator-like gene (*APRR2*) associated with fruit pigment accumulation in melon. Using combined transcriptome and metabolome datasets, Wang et al. [17] successfully constructed a gene-metabolite network and identified that the low expression of *4CL*, *CHS*, and *UFGT* resulted in the decrease in naringenin chalcone, naringenin, and anthocyanin in cucumber.

However, integrated metabolomic and transcriptomic studies for identifying the relationship between melon peel coloration genes and metabolites are very limited. The molecular and metabolic pathways that regulate melon peel coloration are still unclear. In this study, we performed an integrated analysis of the transcriptome and metabolome of three different peel color materials (grey-green peel, dark-green peel, and yellow peel) to explore the molecular mechanisms of peel color formation.

## 2. Results

### 2.1. Identification of DEGs Related to Peel Color

Three cDNA libraries were constructed by sequencing RNA extracted from the peels of W, B, and H samples (Figure 1). The transcriptome sequencing obtained a total of 68.84 Gb clean data. More than 92.08% of the clean data had scores greater than Q30. Interestingly, 87.26% to 89.41% of clean reads were mapped to the reference genome (Appendix A). The correlation coefficients of three biological replicates were more than 0.97 (Figure 2A). A total of 22,090 unique genes were detected among the three peel samples. Using cutoff criteria with a |fold change| ≥ 2 and FDR < 0.05, DEGs from each two-treatment comparison (i.e., W vs. B, W vs. H, B vs. H) were obtained. Hierarchical clustering of the DEGs based on the FPKM values showed the expression pattern of genes in the three comparison groups (W vs. B, W vs. H, and B vs. H). Low and high gene expression levels are represented by green and red bands, respectively (Figure 2B). There were 524 (W vs. B), 5914 (W vs. H), and 6258 (B vs. H) DEGs in the three comparison groups, respectively. Obviously, the two groups W vs. H and B vs. H shared more differentially expressed genes. Two hundred ninety-five up-regulated and 229 down-regulated genes were detected between W and B, 2796 up-regulated and 3118 down-regulated genes were detected between W and H, and 2849 up-regulated and 3409 down-regulated genes were detected between B and H (Appendix A). Venn diagram analysis indicated that 5014 DEGs were common in W vs. H and B vs. H. (Figure 2C). Additionally, these genes differed between the green and yellow peels, implying that they may be responsible for the yellow peel formation.

### 2.2. Functional Analysis of DEGs

In order to identify the function of DEGs in the formation of fruit peel coloration, three categories were classified, including molecular function, cellular component, and biological process, according to GO classifications. In total, 59 GO terms were significantly enriched between W and B, most of which were enriched in the catalytic activity, binding, cell part, metabolic process, and other functional categories (Appendix A, Appendix A). Between W and H, there were 179 GO-annotated DEGs, which were mainly categorized into the cell part, binding, single-organism process, metabolic process, and other functional categories (Figure 3A, Appendix A). One hundred ninety-two GO terms between B and H were enriched in the binding, catalytic activity, cell, cell part, cellular process, and other functional categories (Figure 3B, Appendix A). Then, we used KEGG enrichment analysis of the DEGs to confirm the DEG-associated pathways in the formation of the peel color. Between the W and B libraries, KEGG analysis revealed that beta-alanine metabolism (ko00410), insulin signaling pathway (ko04910), and carotenoid biosynthesis (ko00906) were the top three significantly changed pathways. Between the W and H libraries, KEGG analysis revealed the Fanconi anemia pathway (ko03460), steroid hormone biosynthesis (ko00140), and retinol metabolism (ko00830) were the top three significantly changed pathways. Between the B and H libraries, KEGG analysis revealed Fanconi anemia pathway (ko03460), biosynthesis of secondary metabolites (ko01110), and protein processing in endoplasmic reticulum (ko04141) were the top three significantly changed pathways (Table 1). Clearly, isoflavonoid biosynthesis (ko00943), flavone, and flavonol biosynthesis (ko00944), and carotenoid biosynthesis (ko00906) were significantly changed in W vs. H and B vs. H (Table 1). These pathways may influence fruit peel coloration.

### 2.3. Identified Metabolites Involved in Flavonoid Biosynthesis

We profiled the metabolome analysis of the different peel samples via LC–MS. Abundance lower than 9.00 was not detected. The DEMs were set as fold change ≥ 2 (up-regulation) or ≤ 0.5 (down-regulation). The results of PCA showed that the metabolites were clearly separated among the three samples (Figure 4A). We detected 49, 113, and 129 DEMs in three groups (W vs. B, W vs. H, and B vs. H, respectively) (Figure 4B–D; Appendix A). Twenty-three up-regulated and 26 down-regulated DEMs were identified between W vs. B (Figure 4B), including 39 flavones, three flavonols, one flavanone, one anthocyanin, and other compounds (Appendix A). Eighty-eight up-regulated and 25 down-regulated DEMs were identified between W vs. H (Figure 4C), including 53 flavones, 17 flavanones, 17 flavonols, 7 anthocyanins, 7 isoflavones, and other compounds (Appendix A). In total, 101 up-regulated and 28 down-regulated DEMs were identified between B vs. H (Figure 4D), including 67 flavones, 17 flavanones, 18 flavonols, 8 anthocyanins, 8 isoflavones, and other compounds (Appendix A). Interestingly, we observed a clear distinction between the yellow peel and green peel samples. Moreover, compared to W vs. B, there were more of the same metabolites between W vs. H and B vs. H. In particular, metabolites of flavanones, flavonols, and isoflavones and anthocyanins were present.

### 2.4. KEGG Enrichment Analysis of Significantly Differential Metabolites

The DEMs were mapped to KEGG metabolic pathways. KEGG enrichment analysis showed that the DEMs were mainly involved in four pathways: flavonoid biosynthesis (Ko00941), flavone and flavonol biosynthesis (ko00944), isoflavonoid biosynthesis (Ko00943), and anthocyanin biosynthesis (Ko00942). The four pathways all existed in W vs. H and B vs. H (Figure 5B,C). Additionally, only the flavone and flavonol biosynthesis was enriched in the group W vs. B (Figure 5A). Therefore, the differences between the green peel and yellow peel were mainly caused by the biosynthesis of flavonoid, isoflavonoid, and anthocyanin.

A total of 40 kinds of flavonoids were enriched in four KEGG pathways, including 11 flavones, 10 flavanones, 9 flavonols, 7 isoflavones, and 3 anthocyanins (Table 2). Among these flavonoids, 36 up-accumulated and 4 down-accumulated in the H (Appendix A). Most compounds (flavones, flavanones, flavonols, and isoflavones) were significantly higher in the H sample than in the W and B samples, but the content of anthocyanins was lower in the H sample. Furthermore, the levels of flavonoids accumulation were similar in W and B, but their accumulation pattern was more complicated in H.

We analyzed the flavones in the melon peel. Flavones were detected to be the maximum number of metabolites among metabolites with significant content differences among three melon peel samples. Eleven flavones were identified. All these flavones were significantly higher in W vs. H and B vs. H, ranging from 3.51-fold to 20.18-fold increments. The content of these flavones has no other significant difference in the W and B samples, except luteolin 7-*O*-glucoside and chrysoeriol. The fold change of luteolin 7-*O*-glucoside was significantly decreased (−16.67-fold) in W vs. B (Table 2).

We analyzed the flavanones in the melon peel. Ten kinds of significantly different flavanones were detected. The content of flavanones was considerably higher in H compared with W and B. Moreover, it showed a 10.36-fold to 21.13-fold increase in W vs. H, as well as B vs. H. The levels of these flavanones were not significantly different in W vs. B but showed significant changes in the W vs. H and B vs. H. However, the fold change of homoeriodictyol was 6.82-fold decreased in W vs. B but increased 14.31-fold and 21.13-fold in W vs. H and B vs. H, respectively (Table 2).

We analyzed the flavonols in the melon peel. Nine flavonols in the melon peel were identified. The content of flavonols was higher in the H sample compared with W and B samples. Kaempferol 3-*O*-glucoside, quercetin 3-*O*-glucoside, and kaempferide were the most abundant flavonols in W, B, and H samples, respectively. The content of kaempferide, laricitrin, kaempferol, dihydroquercetin, dihydrokaempferol, kaempferol 3-*O*-galactoside, and syringetin were the same in the W and B samples but exhibited significant changes in the H sample. The content of quercetin 3-*O*-glucoside and kaempferol 3-*O*-glucoside were significantly different in the W, B, and H samples (Table 2).

We analyzed isoflavones and anthocyanins in the melon peel. Seven kinds of significantly different isoflavones were detected in W, B, and H. Among these compounds, the content of formononetin 7-*O*-glucoside was the highest in the W and B samples. Compared to W and B, genistein, genistein 7-*O*-glucoside, calycosin, 6-hydroxydaidzein, 2′-hydroxygenistein, and biochanin A were highly abundant, while formononetin 7-*O*-glucoside was presented in lower content in the H samples. For the anthocyanins, three compounds were identified. All of them were lower in content in the H samples. The fold change of delphinidin 3-*O*-rutinoside, cyanidin 3-*O*-rutinoside, and petunidin 3-*O*-glucoside was 3.17- to 16.75-fold lower in the W vs. H and B vs. H, compared with W vs. B (Table 2).

### 2.5. Modulation of Flavonoid Biosynthesis Genes and Metabolites in Relation to Fruit Peel Coloration

In order to further analyze the relationship between genes and metabolites of flavonoids involved in the coloration of yellow fruit peel, all results of metabolites and genes were combined to establish a network (Figure 6). The results showed that 12 DEGs, 10 DEMs, and their derivatives participated in flavonoid biosynthesis pathway in W, B, and H. The 12 genes with significant differences of flavonoid biosynthesis were as follows: two *4CL* (*MELO3C017009.2*, *MELO3C035535.2*), *F3H* (*MELO3C035771.2*), three *F3′H* (*MELO3C017219.2*, *MELO3C005571.2*, *MELO3C005570.2*), *IFS* (*MELO3C010951.2*), *FNS* (*MELO3C005570.2*), *FLS* (*MELO3C035771.2*), two *ANR* (*MELO3C017061.2*, *MELO3C014584.2*), and *UFGT* (*MELO3C009387.2*) genes. Compared with W and B, there were nine up-regulated and three down-regulated genes in H. A total of 10 DEMs (naringenin chalcone, naringenin, eriodictyol, dihydrokaempferol, kaempferol, kaempferide, dihydroquercetin, cyanidin 3-*O*-rutinoside, delphinidin 3-*O*-rutinoside, petunidin 3-*O*-glucoside), including seven up-accumulated and three down-accumulated ones, were detected. Naringenin is an important compound as the precursor for flavones, isoflavones, flavonols, and anthocyanins [18]. We found that naringenin was significantly different in the peel of green and yellow melon. The content of upstream compounds, naringenin chalcone, naringenin, eriodictyol, dihyrotricetin, dihydrokaempferol, dihydroquercetin, and dihydromyricetin, were all significantly higher in H, compared with W and B. The content of anthocyanins, cyanidin 3-*O*-rutinoside, delphinidin 3-*O*-rutinoside, and petunidin 3-*O*-glucoside were significantly lower in H. It is reasonable to speculate that these genes may be involved in the accumulation of flavonoids (flavonol, flavone, flavanone, isoflavone, and their derivatives) and negative accumulation of anthocyanins (cyanidin 3-*O*-rutinoside, delphinidin 3-*O*-rutinoside, and petunidin 3-*O*-glucoside), which lead to the yellow peel of melon (Figure 6).

Using Pearson’s correlation coefficient [7], we performed comprehensive analyses of transcriptome and metabolome data to investigate the association between genes and metabolites involved in flavonoid biosynthetic pathway. In total, we detected 145 significant correlations (correlation coefficient, R^2^ > 0.9) between six DEGs (*MELO3C035535.2*, *MELO3C017219.2*, *MELO3C005571.2*, *MELO3C009387.2*, *MELO3C014584.2*, *MELO3C035771.2*) and 26 DEMs, including five flavones, seven flavanones, six flavonols, six isoflavones, and two anthocyanins (Appendix A). Each metabolite was correlated with many different genes. Interestingly, flavonol and isoflavone shared the largest number of common genes. This suggested that flavonol and isoflavone might have evolved similar accumulation mechanisms. Of these genes, the correlations between *MELO3C035535.2* and 25 metabolites were greater than 0.97, and *MELO3C035771.2* had the highest correlation score (1) with homoeriodictyol.

## 3. Discussion

Melon shows a large variation in rind color, such as white, yellow, orange, and green [3]. Throughout the development of the fruit, the complicated network of metabolites and genes is dramatically transformed [19]. Fruit peel secondary metabolites, such as pigments, aroma compounds, and tannins, affect fruit appearance [20]. In the current study, we analyzed different metabolites, and flavones, flavanones, flavonols, isoflavones, and anthocyanins were commonly responsible for rind color differences. By analyzing transcriptome data, we uncovered several DEGs related to flavone and flavonol biosynthesis, isoflavonoid biosynthesis, and anthocyanin biosynthesis that are probably involved in rind color development. In addition, a combined transcriptome and metabolome study was conducted to elucidate the mechanism of fruit rind coloration.

Plant metabolomic analysis enables us to study the relationship between metabolites produced by biological processes and plant characteristics [21]. Flavonoids are vital pigments for regulating the rind color of many fruits [11,19,22]. There are six main types of flavonoids in plant tissues: flavones, isoflavones, flavonols, flavanones, flavanols, and anthocyanins. Furthermore, in yellow melons, naringenin chalcone is considered the major flavonoid pigment in rinds [3]. In our study, flavones, flavonols, flavanones, and isoflavones were up-regulated, and anthocyanins were down-regulated in the H samples, compared with W and B samples. Our results perfectly corroborate the conclusions of Wang et al. [23], who proposed that the levels of flavones, flavonols, isoflavones, and anthocyanins determined the green and yellow color of citrus peel. However, they did not analyze the genes associated with peel pigmentation. We used a combination of transcriptome and metabolome data to uncover genes involved in the biosynthesis of flavones, flavonols, isoflavonoids, and anthocyanins, thus searching for valuable information to understand the changes in melon rind color.

Genes that participate in flavonoid biosynthesis and regulation have been reported in melon [15,24]. However, many genes have been discovered and identified via traditional research technology. Transcriptome analysis is considered an important method for studying the expression level, structure, and function of genes in order to uncover phenotypic traits. Thus, the combination of metabolome and transcriptome analyses has increasingly become a practical tool for the mining of genes involved in various metabolic pathways [25].

Using phenylalanine as precursor, flavonoids are synthesized via flavonoid metabolic pathway. First, phenylalanine catabolism is catalyzed by phenylalanine ammonia-lyase (*PAL*), cinnamic acid 4-hydroxylase (*C4H*), and 4 coumarate CoA ligase (*4CL*) to form the starting substrate for flavonoid synthesis, *p*-coumaroyl-CoA. Then, the chalcone precursors for all flavonoids are the condensation of *p*-coumaroyl-COA and three molecules of malonyl-COA by chalcone synthase (*CHS*) [15,26]. Subsequently, chalcone is converted to naringenin by chalcone isomerase (*CHI*). The following reaction catalyzed by flavanone 3-hydroxylase (*F3H*) produces dihydrokaempferol, which is further transformed into dihydroquercetin and dihydromyricetin under the actions of flavonoid 3′-hydroxylase (*F3′H*) and flavonoid 3′,5′-hydroxylase (*F3′5′H*) [27]. Finally, consecutive reactions catalyzed by dihydroflavonol reductase (*DFR*) and anthocyanidin synthase (*ANS*), which can be further methylated, glycosylated, and acylated, lead to the formation of a wide variety of species-specific anthocyanins [28]. Flavone synthase (*FNS*) and flavonol synthase (*FLS*) are responsible for the formation of flavones and flavonols, respectively [29]. Previous studies identified some key structural genes from the flavonoid pathways that regulate coloration. Structural gene *4CL* was down-regulated, which largely explained the reduced accumulation of naringenin chalcone and naringenin [17]. Here, we found that *MELO3C017009.2* and *MELO3C035535.2* may also be involved in the high accumulation of the naringenin chalcone and naringenin content in melon. When the expression of *F3H* gene decreased, the dihydrokaempferol component did not accumulate and the color of petals and fruits had been changed [30,31]. Additionally, the high expression of *F3′H* provides sufficient content of dihydroquercetin [32]. We further identified that the genes *F3H* (*MELO3C035771.2*, Log_2_ fold change values = 13.862, 15.594, respectively) and *F3′H* (*MELO3C017219.2*, Log_2_ fold change values = 2.736, 2.496, respectively; *MELO3C005571.2*, Log_2_ fold change values = 2.825, 2.469, respectively) were highly expressed in H compared to W and B (Appendix A). Moreover, they resulted in dihydrokaempferol and dihydroquercetin accumulation. Overexpression of *IFS* resulted in the accumulation of isoflavones formononetin, daidzein, and genistein glycosides in transgenic plants [27,33,34]. High expression of *FNS* mostly accumulates apigenin and luteolin. The high expression levels of *FLS* observed here are consistent with the essential role in the biosynthesis of the flavonols quercetin and kaempferol, as well as with the critical role of *CnFLS1* in yellow pigmentation in *C. nitidissima* [35]. In agreement with our findings, the genes *IFS* (*MELO3C010951.2*), *FNS* (*MELO3C005570.2*), and *FLS* (*MELO3C035771.2*) were proposed as important structural genes for regulating the pigmentation in melon. Moreover, the genes *ANS* and *UFGT* were found to differentially regulate the anthocyanin accumulation in different melon cultivars [9,26], which was also revealed in this study. Obviously, 12 DEGs associated with isoflavonoid biosynthesis (Ko00943), flavone, and flavonol biosynthesis (Ko00944), and biosynthesis of secondary metabolites (Ko01110) was also related to the formation of peel color. The difference in the color of fruit is mainly due to the differential accumulation of flavonoids and anthocyanins [22]. The present study confirmed these conclusions, and the transcriptome and metabolome analyses highlighted major changes in flavonoid biosynthetic pathway-related genes and metabolites, which may correspond with the changes in peel coloration of melon.

## 4. Materials and Methods

### 4.1. Plant Materials

Three different melon cultivars were used in this study. The fruits were classified into three types according to the peel color: grey-green (W), dark-green (B), and yellow (H) (Figure 1). The fresh fruits were harvested at the maturity stage (35 days after anthesis for W and B, 45 days after anthesis for H) at the Institute of Vegetables and Flowers of the Chinese Academy of Agricultural Sciences in Beijing. Fruits were grown in greenhouse conditions during the spring season of 2018. The fruits were selected for uniformity in color, shape, and size, and they were quickly transported to the laboratory [36]. Fruit peels (0.1 cm thick) were carefully excised with scalpels, collected, immediately frozen in liquid nitrogen, and then stored at −80 °C for subsequent transcriptome sequencing and metabolite extraction. Three independent biological replicates were collected per treatment.

### 4.2. Transcriptome Sequencing

Nine libraries representing the three peel samples and the three replicates were constructed for RNA-Seq. Total RNAs were extracted from W, B, and H samples using an RNA extraction kit (DP441, TIANGEN, Beijing, China). The quantity and quality of mRNAs were measured using a Nano-Drop ultraviolet spectrophotometer (Nanodrop Technologies, Thermo, Waltham, MA, USA) and a Bioanalyzer 2100 System (Agilent Technologies, Santa Clara, CA, USA), respectively. RNA integrity of the samples was determined by 1% agarose gel electrophoresis.

The sequencing libraries were constructed using 3 µg RNA per sample and sequenced using 150 bp paired-end Illumina Hiseq 4000 platform. Briefly, the mRNA was purified using magnetic beads with Oligo (dT). We synthesized first-strand cDNA with random hexamer primer and M-MuLV reverse transcriptase and then second-strand cDNA was synthesized using RNase H and DNA polymerase I. DNA fragments were first repaired at the end. A tail was added, the sequencing connector was connected, and the fragments were purified using the AMPure XP beads to select cDNA fragments of 240 bp in length. PCR was performed, and a cDNA library was constructed by purifying PCR products with the AMPure XP beads. The libraries were sequenced on an Illumina HiSeq platform, and 150 bp paired-end reads were produced.

### 4.3. Transcriptome Data Analysis

Low-quality sequencing reads containing adapter and poly-*N* were removed before downstream analyses. The filtered reads were mapped to the reference genome sequence using HISAT2 software [37]. The gene expression level was calculated by the FPKM (fragments per kilobase of transcript per million fragments mapped) method. Differential expression analyses among the three groups (W vs. B, W vs. H, B vs. H, with three biological replicates per treatment) of colored samples were conducted using the DESeq R package (1.10.1) [38], with adjusted *p*-values. Genes with the following parameters were considered differentially expressed genes (DEGs): |Log_2_fold change| ≥ 1 and false discovery rate (FDR) < 0.05. Gene Ontology (GO) annotation and Kyoto Encyclopedia of Genes and Genomes (KEGG) pathway enrichment analyses of DEGs were applied, employing clusterProfiler software [39].

### 4.4. Metabolite Extraction and Separation

We collected the melon peel samples from grey-green (W), dark-green (B), and yellow (H) fruits. Freeze-dried samples were used in our experiment for metabolome analysis. 100 mg powder was weighted and extracted overnight at 4 °C with 1.0 mL 70% aqueous methanol. Then, the extracts were analyzed via a LC-MS/MS system (HPLC, Shim-pack UFLC SHIMADZU CBM30A system, Shimadzu, Kyoto, Japan; MS, Applied Biosystems 6500 Q TRAP, Applied Biosystems, AB Sciex, Waltham, MA, USA). The analytical conditions were as follows: Waters ACQUITY UPLC HSS T3 C18 column (1.8 µm, 2.1 mm × 100 mm); water (0.04% acetic acid) to acetonitrile (0.04% acetic acid) mobile phase; gradient program: 100:0 (A/B) at 0 min, 5:95 (A/B) at 11.0 min, 5:95 (A/B) at 12.0 min, 95:5 (A/B) from 12.1 min to 15.0 min; flow rate of 0.40 mL/min; column temperature of 40°C; injection volume of 2 μL. The effluent was alternatively connected to an ESI-triple quadrupole-linearion trap (Q TRAP)-MS.

### 4.5. Metabolite Qualification and Quantification

To monitor the repeatability of the analytical process, a quality control sample consisting of a mixture sample extract of every three analytical samples was used during instrumental analysis. Then, raw mass spectrometry (MS) data were imported into Analyst 1.6.3. Metabolites were annotated by public databases, including KNAPSAcK, MassBank, MoToDB, METLIN, and HMDB. The metabolites were quantified using MRM. We used MultiaQuant for peak areas, peak integration, and peak alignment calculations. The differentially expressed metabolites (DEMs) among three peel samples were evaluated using univariate analysis methods, such as fold change (FC) analysis, and multivariate analysis methods, such as partial least squares-discrimination analysis (PLS-DA). The different metabolites were identified with the following parameters: metabolites with VIP ≥ 1 and fold change ≥ 2 or fold change ≤ 0.5 [40]. The different metabolites were subjected to data normalization.

### 4.6. Integrated Metabolome and Transcriptome Analysis

We used principal component analysis (PCA) to compare the trends of metabolites. Enrichment analysis of functions and signaling pathways of the differentially expressed genes was conducted by KEGG database. A *p*-value less than 0.05 was defined as DEGs. Pearson’s correlation coefficients were calculated between the genes and metabolites. On the basis of calculation results, DEGs and DEMs with a coefficient of R^2^ > 0.8 were selected. Then, gene and metabolite datasets were analyzed by canonical correlation analysis (CCA) to reveal melon fruit peel color metabolites and gene molecular interactions, as well as to construct a network.

## 5. Conclusions

In the present study, the regulatory network of fruit peel coloration in melon was carried out by transcriptome and metabolome for the first time. We identified differentially expressed genes associated with fruit rind coloration between yellow and green peel color, among which *4CL*, *F3H*, *F3′H*, *IFS*, *FNS*, *FLS*, *ANS*, and *UFGT* play important roles in flavonoid biosynthetic pathways. Metabolomic analysis indicated that crucial flavone, flavonol, flavanone, isoflavone, and anthocyanins responsible for fruit peel pigmentation showed significantly different expression between yellow and green rind types. In summary, the flavonoids and their correlated genes detected in this study provide a vital metabolic and functional role for flavonoid biosynthetic pathways in melon peel coloration. The findings from our study will benefit molecular breeders in improving the quality of melon fruit.

## Figures and Tables

**Figure 1 molecules-26-02830-f001:**
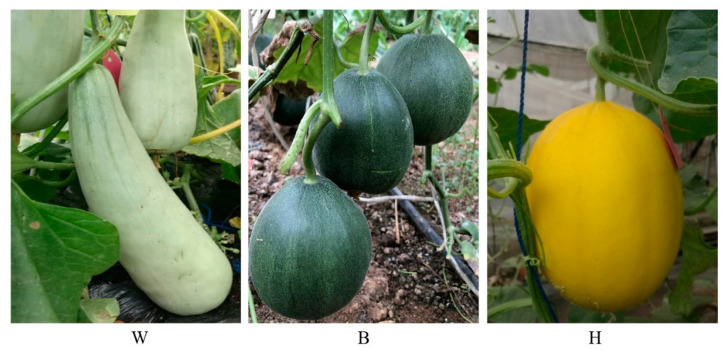
The phenotypes of the fruit peel color: grey-green (**W**), dark-green (**B**), and yellow (**H**).

**Figure 2 molecules-26-02830-f002:**
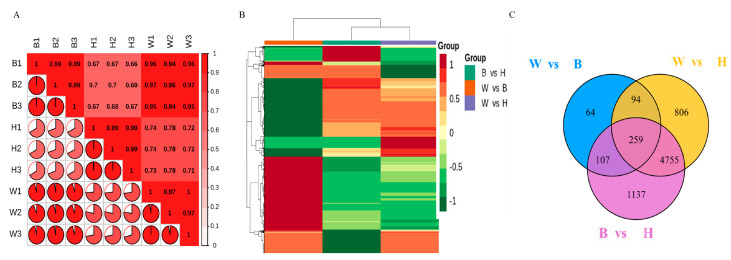
The differentially expressed genes (DEGs) among three comparison groups. (**A**) Sample correlation heat map. (**B**) Hierarchical clustering of DEGs. (**C**) Venn diagram of DEGs.

**Figure 3 molecules-26-02830-f003:**
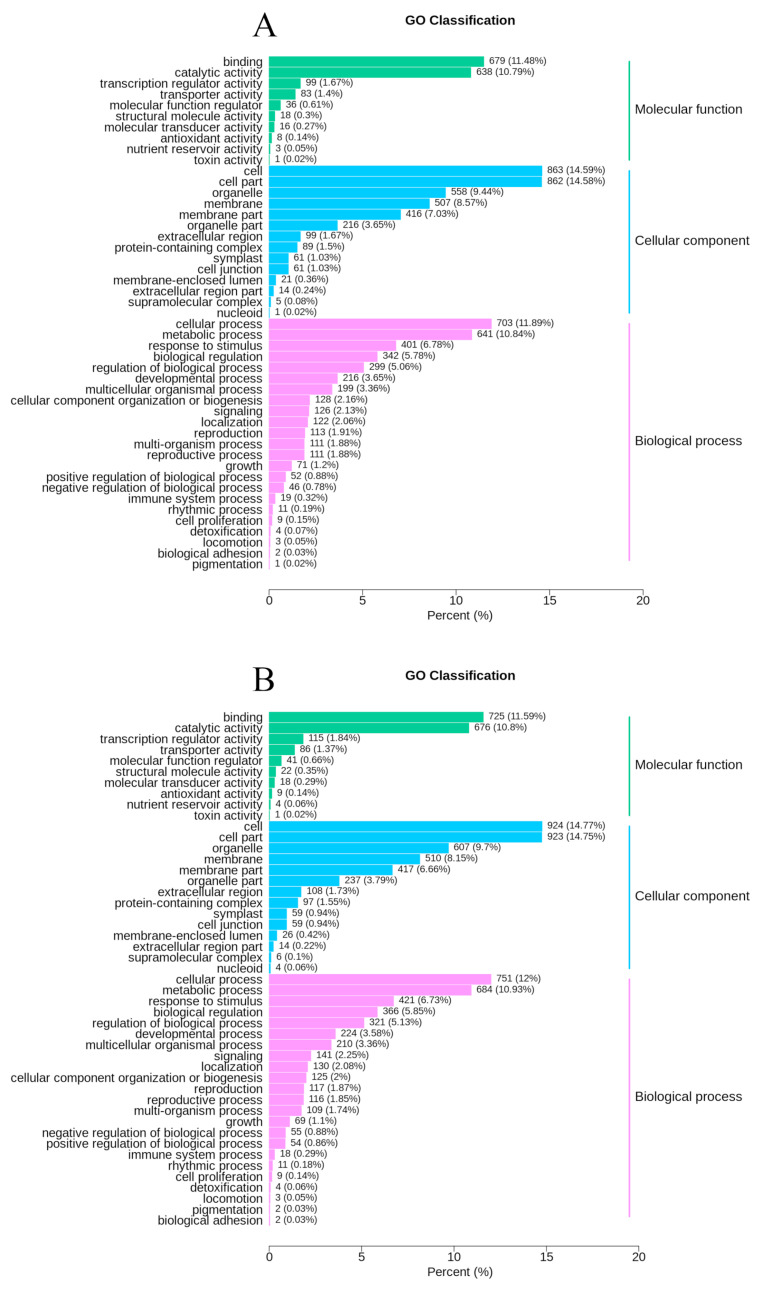
GO analysis of DEGs in (**A**) W vs. H, and (**B**) B vs. H and the number and proportion of DEGs in different functional categories.

**Figure 4 molecules-26-02830-f004:**
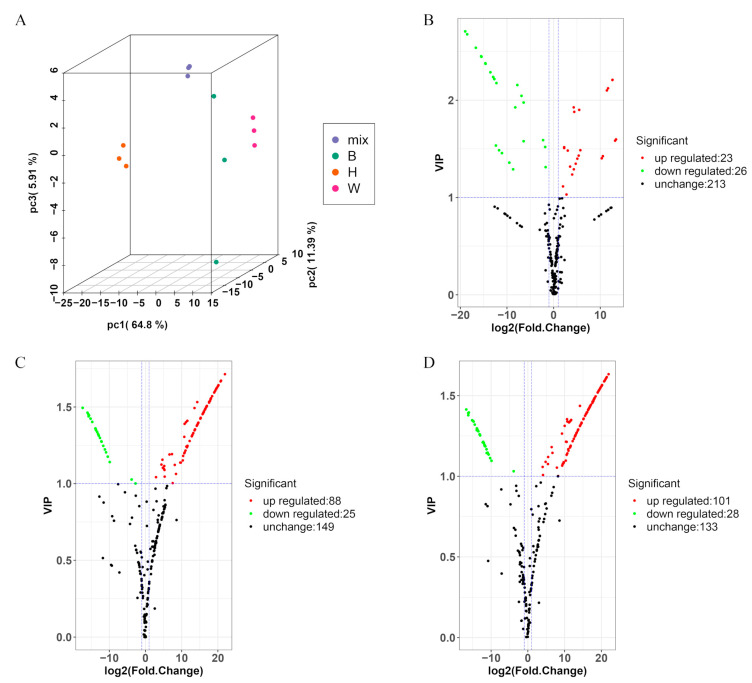
PCA plot, and volcano plots analysis of the DEMs in W vs. B, W vs. H, B vs. H. (**A**) PCA plot. The pale pink, green, and red dots represent the W, B, and H, respectively. Volcano plots analysis of the DEMs in (**B**) W vs. B, (**C**) W vs. H, (**D**) B vs. H. The red color means the metabolites are up-regulated; the green color means the metabolites are down-regulated; the black means the metabolites are unchanged in the two samples.

**Figure 5 molecules-26-02830-f005:**
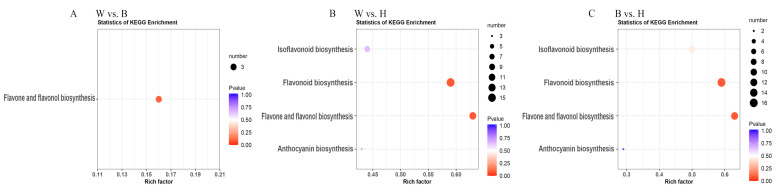
KEGG analysis of the DEMs in (**A**) W vs. B, (**B**) W vs. H, (**C**) B vs. H. KEGG analysis. The *x*-axis represents the richness factor. The color and size of the dots represent *p*-value and the amount of enriched differential metabolites, respectively. Rich factor means the ratio of the number of differential metabolites to the total number of metabolites enriched in a specific category.

**Figure 6 molecules-26-02830-f006:**
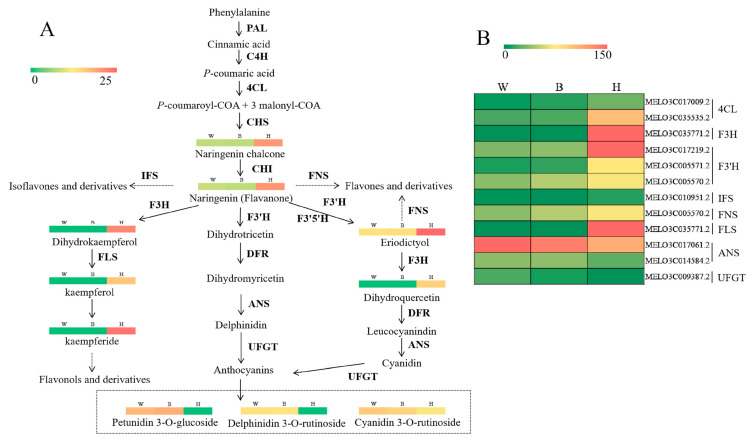
Regulatory network of predicted flavonoid biosynthesis in three different fruit peel samples (grey-green ‘W’, dark-green ‘B’, and yellow ‘H’). (**A**) Color scale from green to red for the heatmap represents the log_2_ value of the metabolite content in W, B, and H. (**B**) Heatmap showing the expression levels of the genes (FPKM value). Red indicates high expression, and green indicates low expression. PAL, phenylalanine ammonia-lyase; C4H, cinnamic acid 4-hydroxylase; 4CL, 4 coumarate CoA ligase; CHS, chalcone synthase; CHI, chalcone isomerase; F3H, flavanone 3-hydroxylase; F3′H, flavonoid 3′-hydroxylase; F3′5′H, flavonoid 3′5′-hydroxylase; FLS, flavonol synthase; FNS, flavonoid synthase; IFS, isoflavone synthase; DFR, dihydroflavonol 4-reductase; ANS, anthocyanidin synthase; UFGT, UDP-flavonoid glucosyl transferase.

**Table 1 molecules-26-02830-t001:** Significantly enriched KEGG pathways among three rind types.

No.	Pathway	Pathway ID	DEGs with Pathway Annotation	All Genes with Pathway Annotation	*p*-Value
W vs. B					
1	beta-alanine metabolism	ko00410	1	1	0.02345415
2	insulin signaling pathway	ko04910	2	12	0.02906519
3	carotenoid biosynthesis	ko00906	1	2	0.04640715
4	mTOR signaling pathway	ko04150	1	2	0.04640715
5	autophagy-animal	ko04140	1	2	0.04640715
6	p53 signaling pathway	ko04115	1	2	0.04640715
7	autophagy-other	ko04136	1	2	0.04640715
W vs. H					
1	fanconi anemia pathway	ko03460	4	7	0.01830560
2	steroid hormone biosynthesis	ko00140	4	7	0.01830560
3	retinol metabolism	ko00830	4	7	0.01830560
4	ras signaling pathway	ko04014	5	11	0.0253581
5	flavone and flavonol biosynthesis	ko00944	2	2	0.0287937
6	isoflavonoid biosynthesis	ko00943	2	2	0.0287937
7	carotenoid biosynthesis	ko00906	2	2	0.0287937
8	tryptophan metabolism	ko00380	4	8	0.03189295
9	ovarian steroidogenesis	ko04913	4	8	0.03189295
10	protein processing in endoplasmic reticulum	ko04141	7	20	0.03807374
B vs. H					
1	fanconi anemia pathway	ko03460	5	7	0.00535022
2	biosynthesis of secondary metabolites	ko01110	24	80	0.02283257
3	protein processing in endoplasmic reticulum	ko04141	8	20	0.03739593
4	plant-pathogen interaction	ko04626	11	31	0.03824971
5	isoflavonoid biosynthesis	ko00943	2	2	0.04330909
6	flavone and flavonol biosynthesis	ko00944	2	2	0.04330909
7	carotenoid biosynthesis	ko00906	2	2	0.04330909

**Table 2 molecules-26-02830-t002:** Differentially identified metabolites in the peel of W, B, and H.

Class	Metabolite Name	Ion Abundance			Log_2_ Fold Change		
		W	B	H	W vs. B	W vs. H	B vs. H
**Flavone**							
	Vitexin 2′′-*O*-beta-l-rhamnoside	9.00	9.00	1.19 × 10^5^	0.00	13.22	13.22
	Luteolin	9.00	9.00	1.17 × 10^6^	0.00	16.99	16.99
	Apigenin 7-*O*-neohesperidoside	9.00	9.00	4.22 × 10^4^	0.00	12.20	12.20
	Tricetin	4.02 × 10^3^	4.26 × 10^3^	3.17 × 10^6^	0.08	9.62	9.54
	Butin	1.31 × 10^3^	1.67 × 10^3^	3.92 × 10^6^	0.35	11.55	11.20
	4,2′,4′,6′-Tetrahydroxychalcone	2.22 × 10^3^	1.43 × 10^3^	3.96 × 10^6^	−0.63	10.80	11.43
	Apigenin	1.47 × 10^4^	1.71 × 10^4^	4.04 × 10^6^	0.22	8.10	7.89
	Apigenin 7-*O*-glucoside	1.76 × 10^4^	3.37 × 10^4^	2.39 × 10^7^	0.93	10.41	9.47
	Chrysoeriol	1.51 × 10^5^	1.93 × 10^4^	5.88 × 10^6^	−2.97	5.28	8.25
	Isovitexin	3.06 × 10^5^	4.33 × 10^5^	5.58 × 10^7^	0.50	7.51	7.01
	Luteolin 7-*O*-glucoside	9.40 × 10^5^	9.00	1.07 × 10^7^	−16.67	3.51	20.18
Flavanone							
	Naringenin 7-*O*-neohesperidoside	9.00	9.00	2.57 × 10^4^	0.00	11.48	11.48
	Naringenin 7-*O*-glucoside	9.00	9.00	4.72 × 10^6^	0.00	19.00	19.00
	Isoliquiritigenin	9.00	9.00	1.16 × 10^5^	0.00	13.66	13.66
	Butein	9.00	9.00	7.95 × 10^5^	0.00	16.43	16.43
	Hesperetin	9.33 × 10^2^	6.09 × 10^2^	1.12 × 10^7^	−0.61	13.55	14.17
	Homoeriodictyol	1.02 × 10^3^	9.00	2.07 × 10^7^	−6.82	14.31	21.13
	Naringenin chalcone	1.29 × 10^3^	1.19 × 10^3^	3.92 × 10^6^	−0.12	11.57	11.69
	Naringenin	1.74 × 10^3^	1.14 × 10^3^	4.03 × 10^6^	−0.61	11.18	11.78
	4′,5,7-Trihydroxyflavanone	5.95 × 10^3^	7.38 × 10^3^	1.50 × 10^7^	0.31	11.30	10.99
	Eriodictyol	3.90 × 10^4^	5.06 × 10^4^	6.65 × 10^7^	0.38	10.74	10.36
Flavonol							
	Kaempferide	9.00	9.00	3.67 × 10^7^	0.00	21.96	21.96
	Laricitrin	9.00	9.00	2.58 × 10^4^	0.00	11.49	11.49
	Kaempferol	9.00	9.00	2.24 × 10^5^	0.00	14.60	14.60
	Dihydroquercetin	9.00	9.00	1.09 × 10^5^	0.00	13.57	13.57
	Dihydrokaempferol	9.00	9.00	1.05 × 10^7^	0.00	20.16	20.16
	Quercetin 3-*O*-glucoside	9.00	1.36 × 10^4^	3.55 × 10^5^	10.57	15.27	4.70
	Kaempferol 3-*O*-galactoside	9.00	9.00	2.31 × 10^6^	0.00	17.97	17.97
	Syringetin	9.00	9.00	5.19 × 10^4^	0.00	3.17	4.94
	Kaempferol 3-*O*-glucoside	1.07 × 10^5^	9.00	1.31 × 10^6^	−13.54	3.60	17.15
Isoflavone							
	Genistein	9.00	9.00	1.78 × 10^6^	0.00	17.59	17.59
	Genistein 7-*O*-Glucoside	9.00	9.00	8.87 × 10^5^	0.00	16.59	16.59
	Calycosin	9.00	9.00	6.12 × 10^5^	0.00	16.05	16.05
	6-Hydroxydaidzein	9.00	9.00	1.13 × 10^6^	0.00	16.93	16.93
	2′-Hydroxygenistein	9.00	9.00	5.52 × 10^5^	0.00	15.91	15.91
	Biochanin A	9.00	9.00	7.63 × 10^3^	0.00	0.40	9.73
	Formononetin 7-*O*-glucoside	5.53 × 10^4^	1.01 × 10^5^	9.00	0.87	−12.59	−13.46
Anthocyanins							
	Delphinidin 3-*O*-rutinoside	4.13 × 10^4^	3.60 × 10^4^	9.00	−0.20	−12.16	−11.96
	Cyanidin 3-*O*-rutinoside	1.58 × 10^5^	1.01 × 10^5^	1.12 × 10^4^	−0.65	−3.82	−3.17
	Petunidin 3-*O*-glucoside	4.97 × 10^5^	9.92 × 10^5^	9.00	1.00	−15.75	−16.75

## Data Availability

The data presented in this study are available on request from the corresponding author.

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
