# Peer review of "Comprehensive Analysis of Transcriptome and Metabolome Reveals the Flavonoid Metabolic Pathway Is Associated with Fruit Peel Coloration of Melon"

_molecules, 2021, doi:10.3390/molecules26092830_

Round 1
Reviewer 1 Report
Manuscript is clear, concisely written, and well organized. Abstract adequately describes the study, principle results and conclusions. Employed experimental methods are adequate, sufficiently clear and complete to allow repetition of the work. Data are properly analyzed and interpreted to support the conclusions. Tables and picture are satisfactory and interpreted correctly. Relevant issues in discussion are adequately discussed. Cited references are appropriate. In my opinion the work is of interest, however it presents fails that should be corrected before publication. See file attached.

Author Response
Dear reviewer,
Thank you so much for the valuable suggestions and comments on our manuscript entitled “Comprehensive analysis of transcriptome and metabolome reveals the flavonoid metabolic pathway is associated with fruit peel coloration of melon”. We revised the manuscript point-by-point carefully. The revised parts were highlighted in red in manuscript. Details are listed as follows:
Point 1: The figure 2, figure 3, and figure 4 are not readable.
Response 1: Thanks very much. The figure 2, figure 3, and figure 4 have been replaced. Please check it. (Page 3, line 98; Page 6, line 143; Page 9, line 236)
Point 2: The mark ‘“’ is redundant in Table 2.
Response 2: Sorry for our carelessness, the mark ‘“’ in Table 2 have been removed. Please check it. (Page 7, line 198)
Point 3: What criteria is used to determine the state of maturity.
Response 3: Thanks for the suggestion. We have added information of maturity. The fruits of W, B, and H were harvested at 35, 35 and 45 days after anthesis. please check it. (Page 11, line 323)

Reviewer 2 Report
First of all, I would like to state to teh authors that the manuscript deals with a very interesting subject and show a new approach to study and elucidate the mechanism involved in fruit coloration. However, while I was reading the manuscript, some questions have risen mainly regarding to the methods used to perform the analysis of the metabolites.
1) Why did the authors perform a GC-MS analysis with the samples and did not show the results obtained?
2) In the topic 4.4 (Metabolite extraction and separation), the authors describe how they performed the preparation of the melon peel extracts from fresh samples for quantification by LC-ESI-MS/MS. However, they do not show some highly important informations regarding the procedure, such as: how much of fresh material was weighted for metabolite extraction; considering the water content can significantly vary in fresh plant material, it is relevant to know if the authors considered the interference of water content in final result of the quantification.
3) Regarding to the metabolite quantification, have the authors considered to use an internal standard in the samples to improved the reliability of the analysis?
In addition to the questions presented above, I would also like to suggest some corrections to the authors:
1) Page 7, line 179: I believe that the authors mean flavonols instead of flavanols.
2) Page 12, line 371: The title of topic 4.5 probably should not has the term quantification twice written (Metabolite quantification and quantification).
Finally, I would like to state the manuscript sounds scientifically relevant in the research area of plant metabolomics, especially applied to a crop of economic importance, but some minor points regarding to the experimental methods used must be clarified before the manuscript acceptance.
Author Response
Dear reviewer,
Thank you so much for the valuable suggestions and comments on our manuscript entitled “Comprehensive analysis of transcriptome and metabolome reveals the flavonoid metabolic pathway is associated with fruit peel coloration of melon”. We revised the manuscript point-by-point carefully. The revised parts were highlighted in red in manuscript. Details are listed as follows:
Point 1: Why did the authors perform a GC-MS analysis with the samples and did not show the results obtained?
Response 1: Sorry for our carelessness, thank you so much. We didn’t perform a GC-MS analysis with the samples. We used LC-MS/MS system (HPLC, Shim-pack UFLC SHI-MADZU CBM30A system, MS, Applied Biosystems 6500 Q TRAP) to analyse samples. We have revised this part. please check it. (Page 12, line 364)
Point 2: In the topic 4.4 (Metabolite extraction and separation), the authors describe how they performed the preparation of the melon peel extracts from fresh samples for quantification by LC-ESI-MS/MS. However, they do not show some highly important informations regarding the procedure, such as: how much of fresh material was weighted for metabolite extraction; considering the water content can significantly vary in fresh plant material, it is relevant to know if the authors considered the interference of water content in final result of the quantification.
Response 2: Thanks for your advice, we have added the information, freeze-dried samples were used in our experiment for metabolome analysis. 100 mg powder was weighted and extracted overnight at 4℃ with 1.0 mL 70% aqueous methanol, please check it. (Page 12, line 362-363)
Point 3: Regarding to the metabolite quantification, have the authors considered to use an internal standard in the samples to improve the reliability of the analysis?
Response 3: Thanks very much, as reviewer suggested that it is better to use an internal standard to improve the analysis. We used LC-MS/MS system (HPLC, Shim-pack UFLC SHI-MADZU CBM30A system, MS, Applied Biosystems 6500 Q TRAP) for the relative quantification of melon metabolites with no internal standard.
Point 4: Page 7, line 179: I believe that the authors mean flavonols instead of flavanols.
Response 4: Sorry for our carelessness. We have changed ‘flavanols’ into ‘flavonols’. Please check it. (Page 7, line 180, line 181, line 183)
Point 5: Page 12, line 371: The title of topic 4.5 probably should not has the term quantification twice written (Metabolite quantification and quantification).
Response 5: Sorry for our carelessness, thank you so much. We have changed the first ‘quantification’ into ‘qualification’ in the title of topic 4.5. Please check it. (Page 12, Line 372)
